# Stereoisomer-Dependent Membrane Association and Capacity for Insulin Delivery Facilitated by Penetratin

**DOI:** 10.3390/pharmaceutics15061672

**Published:** 2023-06-07

**Authors:** Ditlev Birch, Edward J. Sayers, Malene V. Christensen, Arwyn T. Jones, Henrik Franzyk, Hanne M. Nielsen

**Affiliations:** 1Center for Biopharmaceuticals and Biobarriers in Drug Delivery (BioDelivery), Department of Pharmacy, Faculty of Health and Medical Sciences, University of Copenhagen, Universitetsparken 2, 2100 Copenhagen, Denmark; 2Cardiff School of Pharmacy and Pharmaceutical Sciences, Cardiff University, Cardiff CF10 3NB, UK; 3Cancer and Infectious Diseases, Department of Drug Design and Pharmacology, Faculty of Health and Medical Sciences, University of Copenhagen, Universitetsparken 2, 2100 Copenhagen, Denmark

**Keywords:** cell-penetrating peptide, stereoisomers, intracellular uptake, membrane binding, transepithelial permeation, peptide delivery

## Abstract

Cell-penetrating peptides (CPPs), such as penetratin, are often investigated as drug delivery vectors and incorporating d-amino acids, rather than the natural l-forms, to enhance proteolytic stability could improve their delivery efficiency. The present study aimed to compare membrane association, cellular uptake, and delivery capacity for all-l and all-d enantiomers of penetratin (PEN) by using different cell models and cargos. The enantiomers displayed widely different distribution patterns in the examined cell models, and in Caco-2 cells, quenchable membrane binding was evident for d-PEN in addition to vesicular intracellular localization for both enantiomers. The uptake of insulin in Caco-2 cells was equally mediated by the two enantiomers, and while l-PEN did not increase the transepithelial permeation of any of the investigated cargo peptides, d-PEN increased the transepithelial delivery of vancomycin five-fold and approximately four-fold for insulin at an extracellular apical pH of 6.5. Overall, while d-PEN was associated with the plasma membrane to a larger extent and was superior in mediating the transepithelial delivery of hydrophilic peptide cargoes compared to l-PEN across Caco-2 epithelium, no enhanced delivery of the hydrophobic cyclosporin was observed, and intracellular insulin uptake was induced to a similar degree by the two enantiomers.

## 1. Introduction

Cell-penetrating peptides (CPPs) can be efficient vectors for the delivery of associated cargos into the cell cytosol [1,2], as well as across the epithelia [3,4]. Typically, CPPs are employed in tandem with cargo as ionic complexes or as covalent conjugates [5,6]. Al-though the mechanism of CPP internalization is currently not fully understood, it has been reported to involve direct membrane translocation, endocytosis or a combination of both [2,7,8,9]. While there have been no specific structural preferences for either translocation mechanism, it is recognized that several factors affect these processes. These include the CPP structure (both primary [10,11,12] and secondary [13]), concentration [14], cell type [15], peptide-to-cell ratio [16], and cargo type [2]. Since direct translocation delivers the cargo to the cytosol, thereby avoiding the endocytic system, this pathway has the advantage of a rapid introduction of therapeutics to intracellular targets. Direct translocation may be advantageous, e.g., for the treatment of intracellular infections or for inducing necrosis in cancer cells [17]. By comparison, endocytosis may constitute a slower route for delivery, which is primarily caused by the need for endosomal escape following internalization [18,19]. The advantage of endocytosis is that it does not compromise the cell membrane’s integrity, thereby avoiding concurrent cytotoxicity that is related to membrane damage [6,20] as opposed to direct translocation, where this is a risk [21]. In addition to aiming at cargo delivery to intracellular targets, CPPs may be applied as carriers for the delivery of cargo across the epithelia, either by transcytosis, cell membrane perturbation [22], or by affecting the epithelial integrity [3,23].

Penetratin is a very well-studied CPP, and both l- and d-enantiomers (l-PEN and d-PEN, respectively) of this peptide have been examined for their potential to deliver a variety of cargoes to intracellular [6,24,25,26] or transepithelial [5,23,27,28,29,30,31] targets in vitro, as well as in situ [4,32] and in vivo [3,33,34] Nevertheless, their detailed mode of action for enhancing deliveries remains unresolved. While initially believed to translocate across cell membranes by direct translocation [35,36], l-PEN was later, at lower concentrations, demonstrated to internalize by endocytosis [36,37] without compromising the membrane integrity of the examined cells [37]. In contrast, little is known regarding the translocation mechanism of d-PEN into cells, despite having been reported to interact differently in multiple in vitro cell models [25]. The detrimental effects of all-d CPPs have previously been observed at concentrations that were non-toxic for the corresponding l-form [36].

The aim of the present study was to elucidate the mode of action for d-PEN in comparison to l-PEN with respect to cell entry and delivery capacity. The cellular internalization of the carboxyfluorescein (CF)-labeled enantiomers was studied in relevant cell culture models of the gut (Caco-2, IEC-6) and liver (HepG2) as well as for their delivery potential in intracellular and transepithelial delivery. The intracellular delivery of tetramethylrhodamine (TAMRA)-insulin was also studied in Caco-2 cells by confocal laser-scanning microscopy (CLSM) and flow cytometry, while the transepithelial delivery propensity across a well-differentiated Caco-2 cell epithelium was examined for TAMRA-insulin as well as cyclosporin, vancomycin, and mannitol (as a paracellular marker).

## 2. Materials and Methods

### 2.1. Materials

Human colorectal Caco-2 and human liver HepG2 cells were derived from American Type Culture Collection (ATCC, Manassas, VA, USA), and rat small intestine IEC-6 cells were retrieved from the European Collection of Authenticated Cell Cultures (ECACC, Public Health England, Salisbury, UK). (3-(4,5-Dimethylthiazol-2-yl)-5-(3-carboxymethoxyphenyl)-2-(4-sulfophenyl)-2H-tetrazolium (MTS) and phenazine methosulfate (PMS) were from Promega (Madison, WI, USA). Hoechst33342 was from Fisher Scientific (Loughborough, UK). DRAQ7 from Biostatus (Leicestershire, UK), trifluoroacetic acid from Iris Biotech (Marktredwitz, Germany), fetal bovine serum (FBS) was from PAA laboratories (Brøndby, Denmark), Hank’s Balanced Salt Solution (HBSS) from Gibco Life Technologies (Paisley, UK), and N-2-hydroxyethylpiperazine-N’-2-ethanesulfonic acid (HEPES, ≥99.5%) from PanReac AppliChem (Darmstadt, Germany). ^14^C-d-mannitol (57.2 Ci/mmol) and Ultima Gold™ scintillation fluid were obtained from Perkin Elmer (Boston, MA, USA). ^3^H-l-cyclosporin A (20 Ci/mmol) was from American Radiolabeled Chemicals (St. Louis, MO, USA) and ^3^H-l-vancomycin (4.6 Ci/mmol) from ViTrax Company (Placentia, CA, USA). NHS-rhodamine (5/6-carboxy-tetramethyl-rhodamine succinimidyl ester) (TAMRA) was from Thermo Fisher Scientific (Waltham, MA, USA). All other materials were from Sigma-Aldrich (Buchs, Switzerland). Ultrapure water from a Barnstead NanoPure system (Thermo Scientific, Waltham, MA, USA) was used throughout the studies. Unless otherwise specified, all chemicals were of analytical grade or higher.

### 2.2. Methods

#### 2.2.1. Peptide Synthesis, Purification and Labelling

l-Penetratin (l-PEN, RQIKIWFQNRRMKWKK) and d-Penetratin (d-PEN, rqikiwfqnrrmkwkk) as well as their carboxyfluorescein (CF)-labeled analogs (l-PEN_CF_, d-PEN_CF_, respectively) were synthesized by Fmoc-based solid-phase peptide synthesis on an automated CEM Liberty (Matthews, NC, USA) as previously described [38]. Briefly, N-terminal labeling was performed manually overnight in Teflon reactors. The labeling of insulin with TAMRA performed manually by dissolving insulin in water at an acidic pH (pH 1–2) and subsequently increasing the pH to >10 with 1 M Na_2_CO_3_. Hereafter, the solution was mixed with TAMRA (dissolved in DMSO) at a mass ratio of 2:1 insulin:TAMRA and was mixed overnight on a rotary mixer protected from light. The purification of the labeled CPPs, as well as TAMRA-insulin, were performed by RP-HPLC, and purities of >92% were obtained for all conjugates. Conjugate identity was confirmed by MALDI-ToF-MS (Bruker, Microflex LT/SH system). The compounds were freeze-dried, followed by storage at −18 °C.

#### 2.2.2. Cell Culturing

Caco-2, HepG2, and IEC-6 cells were maintained in Corning Costar polystyrene culture flasks (75 cm^2^ or 175 cm^2^ surface area from Sigma-Aldrich) in either Dulbecco’s mo-dified Eagles medium (Caco-2 and IEC-6 cells) or Eagle’s minimum essential medium (HepG2 cells). The media were supplemented with penicillin (100 U/mL), streptomycin (100 µg/mL), l-glutamine (2 mM), non-essential amino acids (1% (*v*/*v*)), and FBS (10% (*v*/*v*) for Caco-2 and HepG2 cells, 5% (*v*/*v*) for IEC-6 cells). HepG2 cells were further supplemented with sodium pyruvate (1 mM) and IEC-6 cells with insulin (5 µg/mL). The cells were kept in a humidified incubator at 37 °C and 5% CO_2_ and subcultured weekly at approximately 90% confluency by trypsinization. The cells used for flow cytometry were seeded in 24-well plates (1.90 cm^2^ surface area, Corning Costar, Sigma-Aldrich, Brøndby, Denmark) or 12-well plates (3.80 cm^2^ surface area) at densities of 181,800 cells/cm^2^, 69,700 cells/cm^2^, and 48,500 cells/cm^2^ for Caco-2, HepG2 cells, and IEC-6 cells, respectively, and were kept for 24 h in a humidified incubator (37 °C and 5% CO_2_). The cells used for microscopy were seeded in flat-bottomed, glass-bottomed microchambers (1 cm^2^ surface area, Ibidi, Planegg/Martinsried, Germany) at densities of 89,300 cells/cm^2^, 15,000 cells/cm^2^, and 10,000 cells/cm^2^ for Caco-2, HepG2, and IEC-6, respectively, and were kept for 72 h in a humidified incubator. Plastic-bottom plates for Caco-2 cells were coated using collagen (8.93 µg/cm^2^) prior to seeding. The variations in cell densities and incubation length were necessary to ensure uniform confluence (approx. 90%). For permeation studies, Caco-2 cells were seeded at a density of 1 × 10^5^ cells/filter insert (pore size: 0.4 µm, area: 1.12 cm^2^; Corning) and were cultured for 18–20 days to obtain well-differentiated Caco-2 cell monolayers. All experiments were performed in at least two passages of cells, over 16, 6, and 9 passages of Caco-2, HepG2 and IEC-6 cells, respectively.

#### 2.2.3. Intracellular Distribution and Uptake of CF-Labelled PEN Enantiomers and TAMRA-Insulin Evaluated by Confocal Microscopy

Cells seeded in microchambers were washed with hHBSS (10 mM HEPES, 0.05% (*v*/*v*) bovine serum albumin (BSA) in HBSS at pH 7.4) and preheated to 37 °C. The uptake and distribution of PEN-enantiomers were investigated in Caco-2 cells, HepG2 cells, and IEC-6 cells by the addition of 250 µL l-PEN_CF_ or d-PEN_CF_ at 20 µM in hHBSS (pH 7.4). Additionally, the uptake of TAMRA-insulin:PEN mixtures was investigated in Caco-2 cells at a 1:4 molar (5 µM TAMRA-insulin, 20 µM PEN enantiomer) using the same buffer as described above. The cells were kept for 1 h on a shaking table (37 °C, 50 rpm) and were washed with hHBSS and stained with Hoechst 33,342 and DRAQ7 (5 µg/mL and 0.3% (*v*/*v*), respectively) for 10 min. Hereafter, the cells were kept in hHBSS prior to imaging with an LSM 780 Zeiss inverted confocal microscope (Carl Zeiss, Oberkochen, Germany) or a Leica SP5 inverted laser scanning confocal microscope (Leica Microsystems, Milton Keynes, UK), equipped with 20× air or 63× oil-immersion objectives. Fluorescence was recorded for Hoechst 33342, CF-labelled PEN, TAMRA-insulin, and DRAQ7 using excitation lasers at 405 nm, 488 nm, 543 nm, and 647 nm, respectively. Pulse/chase experiments of l-PEN_CF_ and d-PEN_CF_ (20 µM in hHBSS, pH 7.4) were performed as above, with the addition that the cells were incubated (37 °C, 5% CO_2_) in hHBSS (pH 7.4) after the first image acquisition (t = 0). Following 1 h and 4 h of the chase, the cells were imaged using the same settings. All images were analyzed using Fiji ImageJ [39].

#### 2.2.4. Cellular Uptake of CF-Labelled PEN Enantiomers and TAMRA-Insulin Evaluated by Flow Cytometry

The cellular uptake of CF-labelled l-PEN and d-PEN into Caco-2 cells, HepG2 cells, and IEC-6 cells, and the PEN-mediated delivery of TAMRA-insulin were evaluated by flow cytometry. Cells were washed twice with hHBSS, followed by the addition of test solutions containing 1–20 µM l-PEN_CF_ or d-PEN_CF_, TAMRA-insulin (5 µM), or TAMRA-insulin:PEN mixtures (5 µM:20 µM), prepared in hHBSS (pH 7.4). The controls were incubated with a buffer without CPP or insulin. The cells were kept on a shaking table (37 °C, 50 rpm) for 1 h. Hereafter, the test solutions were removed, and the cells were washed twice with cold phosphate-buffered saline (4 °C) and trypsinized for 10–15 min. Trypsinization was terminated by the addition of excess cold hHBSS (4 °C) containing 10% (*v*/*v*) FBS (hHBSS/FBS). The cells were then collected and centrifuged at 4000 rpm (1076× *g*, 4 °C). The supernatant was removed, and the cells were washed once with hHBSS/FBS and centrifuged again at the same conditions before suspension in hHBSS/FBS containing 0.3% (*v*/*v*) DRAQ7 as a marker of the plasma membrane’s integrity. To quench extracellular plasma membrane-associated fluorescence, a parallel experiment was performed using the same test solutions, with the addition of trypan blue (TB) at a final concentration of 0.1% (*v*/*v)* just prior to analysis. The mean cellular fluorescence intensity of CF-labelled PEN, TAMRA-insulin, and DRAQ7 was determined from intensities in 10,000 viable cells by flow cytometry using a Gallios flow cytometer (Beckman Coulter, Fullerton, CA, USA) with gating for forward and sideways scattering.

#### 2.2.5. Transepithelial Permeation of Cargoes Facilitated by l-PEN and d-PEN

The permeation of the different cargoes, TAMRA-insulin, vancomycin, and cyclosporin, was evaluated across well-differentiated Caco-2 epithelium in the presence of l-PEN and d-PEN. Cell monolayers were washed twice with hHBSS (37 °C) and equilibrated for 20 min prior to the measurement of the transepithelial electrical resistance (TEER) using a resistance chamber connected to a voltmeter (Endohm and EVOM, respectively, both from World Precision Instruments, Sarasota, FL, USA). Monolayers displaying an initial TEER below 200 Ω × cm^2^ were discarded. Thereafter, the wash medium was replaced with 350 µL of mHBSS (pH 6.5, 37 °C) and 1000 µL of hHBSS (pH 7.4, 37 °C) on the apical and basolateral sides, respectively. The cell monolayers were kept on a shaking table (37 °C, 50 rpm) for 5 min before replacing the buffers on the apical side with 350 µL test solution (10 µM:40 µM cargo:PEN in mHBSS at pH 5 or 6.5). Monolayers exposed to 10 µM of cargo in the absence of PEN were used as controls. Cell monolayers exposed to vancomycin or cyclosporin were spiked with ^3^H-vancomycin or ^3^H-cyclosporin for a final activity of 1 µCi/mL on the apical side. In addition, apical volumes on all cell monolayers were spiked with ^14^C-d-mannitol (final activity of 1 µCi/mL on the apical side) for the simultaneous measurement of the paracellular marker. These cells were kept on a shaking table (37 °C, 50 rpm) for 4 h with the collection of samples from the basolateral side at 15–30 min intervals. The collected sample volume was replaced with hHBSS (37 °C, pH 7.4). After the final sampling, TEER was measured using the previously described equilibration procedure. Afterward, the viability of the cell in the monolayers was determined by using 350 µL of the MTS/PMS buffer (240 µg/mL MTS and 2.4 µg/mL PMS in hHBSS pH 7.4) and 1000 µL hHBSS on the apical and basolateral side, respectively. After 1 h, the MTS/PMS buffer was collected from the apical side and analyzed at 492 nm using a plate reader (FLUOStar OPTIMA, BMG LABTECH, Ortenberg, Germany). The fluorescence of TAMRA-insulin samples was analyzed at a wavelength of 545/590 nm (λ_ex_/λ_em_) on the same plate reader. Scintillation fluid (2 mL) was mixed with each sample that was subsequently quantified by liquid scintillation (Perkin-Elmer Tri-Carb 2910 TR, Perkin Elmer, Boston, MA, USA).

#### 2.2.6. Data Analysis

All statistical analysis was performed in GraphPad Prism 6 (GraphPad Software Inc., La Jolla, CA, USA) using a one-way ANOVA combined with Tukey’s multiple comparison analysis. The calculation of P_app_ from the permeation experiments was calculated as described by Artursson et al. [40]. N represents the number of individual passages, and n represents the repeats within one passage. In the cases where we compared data from less than three passages, the technical replicates in each passage were at least three.

## 3. Results

### 3.1. Effect of Stereoisomerism on the Uptake and Cellular Distribution of l- and d-Penetratin

To investigate the translocation of both PEN enantiomers across the cell membranes, three cell culture models were incubated with l-PEN_CF_ and d-PEN_CF_ for 1 h while monitoring the uptake by CLSM. All cell models incubated with 20 µM l-PEN_CF_ displayed vesicular staining of the cytoplasm (Figure 1), which was consistent with endocytic uptake. Additionally, l-PEN_CF_ strongly stained the plasma membrane of Caco-2 cells, whilst IEC-6 cells demonstrated diffuse cytosolic staining. Caco-2 cells incubated with d-PEN_CF_ also displayed vesicular staining; however, it was prominent that clusters of fluorescence, which did not resemble any typical cytosolic pattern, were observed in this case. A similar phenomenon was seen in the other two cell lines, whereas d-PEN_CF_ labeling was not prominent on the plasma membrane in any cell model but appeared as aggregates in all cell lines. The white arrows depicting DRAQ7 clearly show that in HepG2 cells and, to a lesser extent, in IEC-6 cells, 20 µM d-PEN_CF_ caused plasma membrane porosity under these experimental conditions.

To further investigate the internalization of l-PEN_CF_ and d-PEN_CF_, they were incubated with Caco-2 cells and subjected to a 0 h, 1 h, or 4 h endocytic chase study to follow the intracellular distribution of internalized CPP (Figure 2). The 0 h chase time point was identical to previous observations (Figure 1); however, after 1 h of the chase, it was observed that the level of plasma membrane-associated l-PEN_CF_ decreased. An enrichment of fluorescence in the perinuclear region of cells was pronounced after 4 h of chase upon l-PEN_CF_ incubation. A very different spatiotemporal fluorescence profile was shown for d-PEN_CF_, showing the aggregation of fluorescence at all time points, which is consistent with the observations presented in Figure 1.

Caco-2 cells and HepG2 cells were also subjected to analysis by flow cytometry to provide a semi-quantitative evaluation of the cellular uptake of l-PEN_CF_ and d-PEN_CF_ after incubation with the 1–20 µM peptide. The enantiomers of PEN_CF_ were taken up at similar levels in the cells when incubated at concentrations in the range of 5–20 µM (Figure 3). A parallel experiment was performed under the same conditions as previously described, with the exception that Trypan Blue (0.1% (*v*/*v*) final concentration) was added as an extracellular quencher to the samples just prior to the analysis by flow cytometry. For data analysis, the relative mean fluorescence intensity (MFI) was normalized to the fluorescence of cells incubated with 20 µM l-PEN_CF_ without Trypan Blue treatment. Under those conditions, the cells exposed to d-PEN_CF_ exhibited an >95% reduction in fluore-scence at any concentration. By comparison, cells exposed to l-PEN_CF_ displayed an approximately 50% decrease in the detected fluorescence at any concentration. HepG2 cells exposed to d-PEN_CF_ at concentrations above 10 µM displayed morphological aberrations, and these were omitted from further analysis.

### 3.2. l-PEN and d-PEN as Carrier Peptides for Transepithelial Peptide Delivery

The delivery propensities of PEN enantiomers were investigated by performing permeation assays across well-differentiated Caco-2 cell monolayers. Cargoes displaying distinct physicochemical properties (Table 1) were employed to assess the PEN delivery efficiency for different structures. The delivery was assessed at different pH values resembling the variation that could be encountered in the small intestines, as the delivery efficiency of l-PEN was previously shown to depend on pH [27,31].

#### 3.2.1. Intracellular Delivery of TAMRA-Insulin

The uptake and distribution of TAMRA-insulin mediated by PEN-enantiomers were examined in undifferentiated Caco-2 cells by using CLSM. These cells displayed the uptake of the TAMRA-insulin conjugate at both investigated pH values (pH 6.5 and 7.4), independent of the enantiomer present. In contrast, for the control cells incubated with TAMRA-insulin alone at pH 7.4, uptake was absent (Figure 4). At pH 6.5, there was a relatively strong localization of fluorescence at the plasma membrane with some punctate intracellular fluorescence, whereas cells exposed to TAMRA-insulin at pH 7.4 displayed only intracellular vesicular staining. The apparent cellular uptake was higher at pH 6.5 compared to that seen at pH 7.4, albeit it was not possible to clearly distinguish the membrane associated with internalized TAMRA-insulin at pH 6.5.

Flow cytometry analysis showed that l-PEN and d-PEN both increased the cellular uptake of TAMRA-insulin compared to the control cells at pH 6.5 and 7.4 (Figure 5), and the uptake of TAMRA-insulin was higher for the d-PEN-mediated delivery compared to that mediated by l-PEN at both pH 6.5 and 7.4. However, it was evident that quenching extracellular fluorescence with Trypan Blue only reduced the mean cell fluorescence intensity, indicative of TAMRA-insulin uptake, when this was co-incubated with d-PEN and not when it was co-incubated with l-PEN (Table 2).

#### 3.2.2. PEN-Mediated Permeation of Cargo across Caco-2 Cell Epithelium

The capacity for l-PEN and d-PEN to mediate the delivery across the Caco-2 cell epithelium was investigated for cargoes with different physicochemical properties and, thus, different permeation properties. It was observed that incubation with both enantiomers combined with any of the cargoes led to a reduction (by ~25%) in the metabolic activity of Caco-2 cells at pH 5, and a similar reduction was observed for the d-PEN and cargo at pH 6.5 compared to the level for the buffer control cells (Figure 6A). By contrast, l-PEN elicited no metabolic activity changes at pH 6.5. In addition, cells exposed to d-PEN displayed a reduction of ~50–75% in TEER (Figure 6B) and an increase in the P_app_ of mannitol (Figure 7) at both investigated pH values.

d-PEN mediated a 4-5-fold increased permeation of TAMRA-insulin at both pH 5.0 and 6.5 and for vancomycin at pH 5.0 compared to the permeation seen in the buffer alone. However, l-PEN elicited only a small increase in P_app_ for both cargoes at both pH values, and it did not differ from that observed in the control experiments without l-PEN. The permeation of cyclosporin was unaffected by the presence of either peptide enantiomer compared to controls at the respective pH values. The permeation of the paracellular marker mannitol increased significantly in the presence of d-PEN but not when combined with l-PEN.

## 4. Discussion

### 4.1. Enantiomers of PEN Interact Differently with Cell Membranes

The interactions between CPPs and synthetic and natural cell membranes have been extensively investigated [35]. The current consensus is that there is no universal cellular uptake mechanism for CPPs and that they internalize into cells by direct translocation, endocytosis, or via a combination of these processes [2,7,9]. The present study highlights that stereochemistry and cell type determine the internalization patterns, membrane binding, and cargo delivery. It was previously reported that different cell models exhibited different CPP internalization patterns, as shown by vesicular or cytosolic staining [6,20], corroborating our data.

In the current study, l-PEN_CF_ displayed vesicular staining in all the investigated cell models, along with distinct membrane staining in the Caco-2 cell model (Figure 1). The vesicular staining was consistent with the uptake by endocytosis, as previously reported for both l-PEN [6] and fluorescently labeled l-PEN analogs [41]. Cytosolic labeling was very prominent in IEC-6 cells incubated with l-PEN_CF_ and also had evidence of membrane permeabilization (DRAQ7). For the same experiment with d-PEN_CF,_ there was some evidence of cytosolic labeling but also of the loss of plasma membrane integrity. The loss of membrane integrity was also noted in HepG2 cells incubated with the d-form. These observations are consistent with a study by Duchardt et al. [6], who observed a similar pattern in HeLa cells and also reported that direct translocation was only observed for CPP concentrations above a certain threshold and that it appeared to depend on both cell density and cell type. In the present study, the cell densities were comparable in terms of confluence at the time of use, and the concentrations of the enantiomers, when tested, were the same. Interestingly, the initial pronounced Caco-2 plasma membrane association of l-PEN_CF_ was replaced by predominantly vesicular staining upon 1 h and 4 h chases (Figure 2), inferring endocytic trafficking of l-PEN_CF_ from the plasma membrane. When applied in the same concentration as l-PEN_CF_, d-PEN_CF_ displayed clustering in all the examined cell models. These clusters were likely localized at the cell membrane as the fluorescence was fully quenched by the addition of extracellular Trypan Blue (Figure 3). Further, for HepG2 and IEC-6 cells, exposure to d-PEN_CF_ resulted in pronounced membrane damage since a significant proportion of the cells displayed DRAQ7 in the staining of the nuclei (Figure 1). These cell models also displayed cytosolic staining and an absence of vesicular staining with d-PEN_CF_, suggesting that this CPP was internalized by increasing the fluidity of the cell membrane, thereby bypassing the endocytic pathway. This may, in part, be explained by the increased stability of the all-d enantiomer compared to the all-l enantiomer when exposed to cells [29], thereby allowing for longer residence at the membrane. This is also consistent with a study by Watkins et al. [42], who reported cytosolic staining upon the exposure of HeLa cells to R_8_-PAD at concentrations that also induced the uptake of DRAQ7.

In another study by Verdurmen et al. [26], d-enantiomers of CPPs, including PEN, were reported to display reduced uptake in HeLa cells and MC57 cells; these were identified as having high levels of plasma membrane heparan sulfate proteoglycans (HSPG). By comparison, Jurkat cells, lacking plasma membrane HSPG displayed no reduced uptake. However, HSPGs were previously reported not to interact with non-labeled l-PEN [43]. In addition, HSPGs were investigated together with CPPs for their potential role in peptide clustering [44]. In line with this, the observation that d-PEN_CF_ exhibited significant aggregation suggests that there was a pronounced interaction between d-PEN_CF_ and HSPGs on the cell membrane compared to that of l-PEN_CF_. d-PEN_CF_ also displayed aggregation when examined in Caco-2 cells and was accompanied by structures indicative of endocytic vesicles. Both Caco-2 and HepG2 cells were examined by flow cytometry and were found to display the pronounced quenching of fluorescence in response to the addition of extracellular Trypan Blue, suggesting that most of the detected l-PEN_CF_ and especially d-PEN_CF_ was membrane-bound rather than internalized. Further, membrane-bound d-PEN_CF_ was not found to traffic intracellularly following 1 h and 4 h chases, as opposed to that seen for l-PEN_CF_. This difference may have arisen from a combination of the increased proteolytic stability of all-d enantiomers [29,33] and the different binding of the PEN enantiomers to surface-bound HSPGs.

### 4.2. Intracellular Delivery and Distribution of Insulin Are Mediated by Both l-PEN and d-PEN

In the current study, TAMRA-insulin was used as a model cargo for delivery to and through Caco-2 cells. Insulin is a thoroughly studied peptide, which was previously found to permeate intestinal epithelium when co-administered with l-PEN [3,28]. At pH 6.5, TAMRA-insulin was associated with the plasma membrane in the presence of 20 µM l-PEN or d-PEN as well as in the control cells. However, this association appeared slightly more pronounced in cells exposed to d-PEN compared to cells treated with l-PEN (Figure 4). The flow cytometry studies demonstrated a reduction in the fluorescence of TAMRA-insulin in cells upon the quenching of extracellular fluorescence by Trypan Blue as well as in cells incubated with d-PEN but not with l-PEN (Table 2). While some vesicular staining was seen at pH 6.5, these observations inferred that a large part of the detected TAMRA-insulin was not internalized but was instead localized to the plasma membrane at pH 6.5. However, at pH 7.4, pronounced vesicular localization without membrane staining was observed when TAMRA-insulin was applied together with any of the PEN enantiomers, whereas no staining was evident in the insulin-only control. It appears likely that TAMRA-insulin was internalized by endocytosis, potentially as a result of complexation between insulin and PEN, which was especially prominent at pH 7.4 due to their opposite net charges at this pH [28]. No signs of negative effects, e.g., in terms of the decreased viability of the cells, were observed upon exposure to 20 µM l-PEN or d-PEN, which agrees with previously reported data [29].

### 4.3. Transepithelial Delivery of Hydrophilic Peptide Cargoes Is Mediated by Both Enantiomers, yet Preferentially by d-PEN

In the current study, it was observed that d-PEN interacted with Caco-2 cell monolayers through a mechanism, which at the investigated concentration affected the integrity of the epithelium as reflected in reduced TEER values and lowered metabolic activity after exposure for 4 h. Previously, we reported that the exposure of a Caco-2 cell monolayer to 50 µM d-PEN did not result in morphologies different from that seen for the buffer control as evaluated by transmission electron micrographs [29], while 60 µM l-PEN did not reduce the TEER of the monolayer [3]. An exact mechanism for the observed effects in our study of d-PEN in reducing the viability and TEER cannot be proposed, but the increased stability of d-PEN during incubation with the Caco-2 cell monolayers compared to that of l-PEN [29] may contribute. Further, the different membrane binding behaviors, as reported here for the CF-labelled versions of the enantiomers, may well be reflected in different effects on the TEER of the Caco-2 monolayer.

The widening of the paracellular space could allow for the increased paracellular diffusion of hydrophilic cargoes such as vancomycin, insulin, and mannitol. Recently, Maher et al. [30] reviewed several types of permeation enhancers and highlighted that many traditional permeation enhancers interact with the plasma membrane in a way that confers decreased TEER values. This indirect effect is for some enhancers hypothesized to be due to membrane perturbation, which in turn causes the instability of intracellular levels of Ca^2+^ or intracellular kinases, regulating the activity of tight junction proteins [30]. Additionally, studies on the interaction of the cationic AMP melittin with epithelium have reported that such interactions increased the paracellular diffusion of several compounds [22]. This effect was ascribed to the ability of melittin to elicit membrane perturbation, which in turn affected the tightness of the epithelium. In the present study, no signs of membrane perturbation were evident in undifferentiated Caco-2 cells (concluded from the lack of staining of nuclei with DRAQ7 included in the experiment); hence, membrane interaction without perturbation may likely account for the decreased TEER in the polarized monolayer. Membrane interactions have been proposed to be the primary mechanism for fatty acid-based permeation enhancers, which were found to induce similar effects on Caco-2 monolayers as observed for d-PEN in the current study. Here, Caco-2 cell epithelium, when exposed to 40 µM l-PEN elicited no increase in epithelial permeability as neither a decrease in the TEER (Figure 6B) nor increased permeation of the cargo (Figure 7) was observed. A previous report partly supported this by concluding that 20 µM l-PEN did not decrease TEER at either pH 5, 6.5, or 7.4, although it increased insulin permeation at pH 5 [28]. In contrast, a study by Kamei et al. [45] reported that both 60 µM l-PEN and d-PEN were found to increase the permeation of insulin across Caco-2 cell monolayers with an increasing pH from pH 5, 6, 7, to 8, while the TEER was unaffected upon exposure to any of the enantiomers and pH values. Neither l-PEN nor d-PEN increased the permeation of the hydrophobic peptide cyclosporin (Figure 7), supporting a mechanism that may include membrane interactions, while d-PEN had some effect on the TEER, but not to a level leading to the measurably increased permeation of a hydrophobic compound, which was expected to permeate primarily via the transcellular pathway. Overall, the increased proteolytic stability of the all-d stereoisomer, as demonstrated both on the Caco-2 cell monolayer [29] as well as in rat intestinal fluid [33], may indeed explain the improved delivery capacity of the CPP compared to the all-l stereoisomer. It is well-known [46] that one of the strategies to increase the stability of peptides in the gut is to introduce d-amino acids in the molecule, and this could naturally also be applied for cell-penetrating peptides when applied as carriers for oral drug delivery. Thus, the fact that d-PEN displayed improved delivery capacity for hydrophilic cargoes supports further studies on this stereoisomer as a carrier for therapeutic peptide delivery.

## 5. Conclusions

Overall, the stereochemistry of PEN influences the transepithelial delivery propensity but not the cellular uptake. The enantiomers were found to induce similar levels of cellular uptake for insulin in Caco-2 cells, and in particular, d-PEN induced noticeable membrane binding of insulin. The l-PEN enantiomer labelled with CF displayed a clear plasma membrane association prior to its uptake within vesicular structures, whereas the corresponding CF-labeled d-PEN at similar concentrations displayed aggregation at the plasma membrane. While l-PEN did not increase the transepithelial permeation of the studied cargo peptides at the investigated concentration, d-PEN increased the transepithelial delivery of the glycopeptide vancomycin five-fold at pH 5 and nearly four-fold for insulin irrespective of the pH. Moreover, the increased permeation of the paracellular marker pointed toward the increased paracellular permeation of the cargo peptides. By contrast, neither enantiomer was found to increase the transepithelial delivery of the hydrophobic cyclosporin. Thus, the mechanism of d-PEN may include a promoted cargo permeation via other pathways than those involved in l-PEN, which potentially is related to the significant membrane binding of d-PEN. Overall, these studies have implications for the use of this CPP and different CPP enantiomers for the oral delivery of macromolecular therapeutics.

## Figures and Tables

**Figure 1 pharmaceutics-15-01672-f001:**
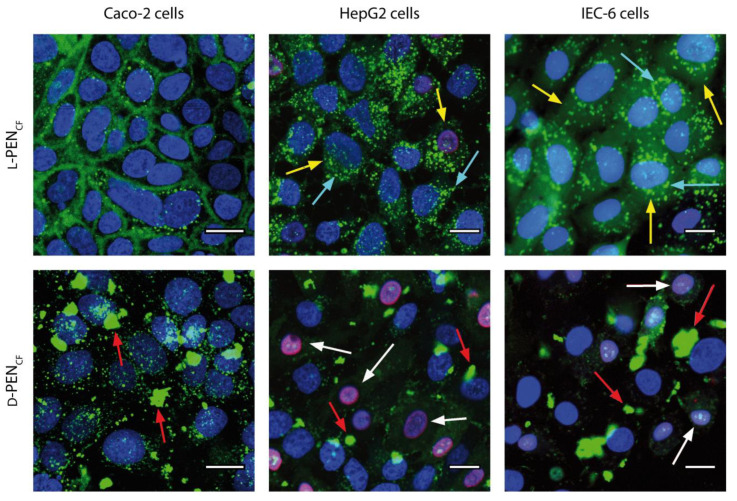
Cellular uptake of l-PEN_CF_ and d-PEN_CF_ in Caco-2 cells, IEC-6 cells, and HepG2 cells. The examined cells were incubated with l-PEN_CF_ or d-PEN_CF_ (20 µM, green) in hHBSS (pH 7.4) for 1 h, followed by the staining of nuclei with Hoechst 33,342 (blue) and DRAQ7 (magenta, white arrows). Vesicular staining (cyan arrows), cytosolic staining (yellow arrows), and CPP clusters (red arrows) are indicated by arrows. Micrographs are representative of two passages of cells. Scale bar = 20 µm.

**Figure 2 pharmaceutics-15-01672-f002:**
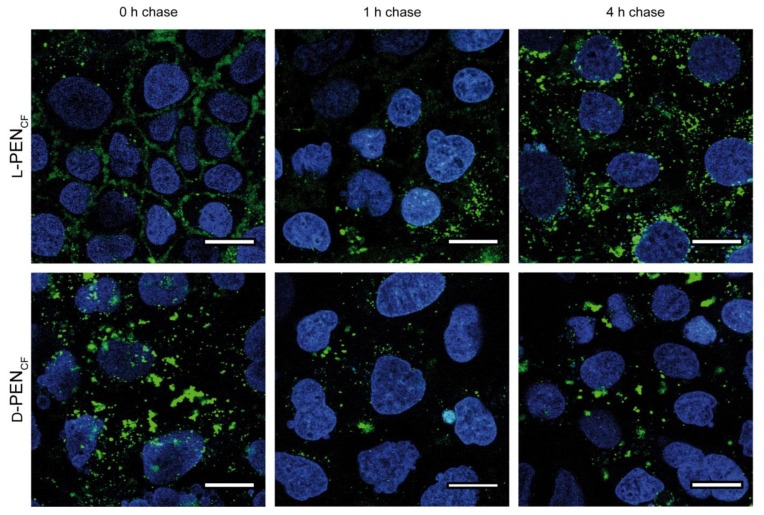
Cellular uptake of l-PEN_CF_ and d-PEN_CF_ in Caco-2 cells following 0 h, 1 h, and 4 h chase in hHBSS (pH 7.4). Caco-2 cells were incubated with 20 µM peptide (green) in hHBSS (pH 7.4) for 1 h at 37 °C. The cells were washed, and nuclei were stained with Hoechst 33,342 (blue) prior to imaging. The images are representative of the same passage of cells, and the experiment was repeated in two different passages of cells. Scale bar 20 µm.

**Figure 3 pharmaceutics-15-01672-f003:**
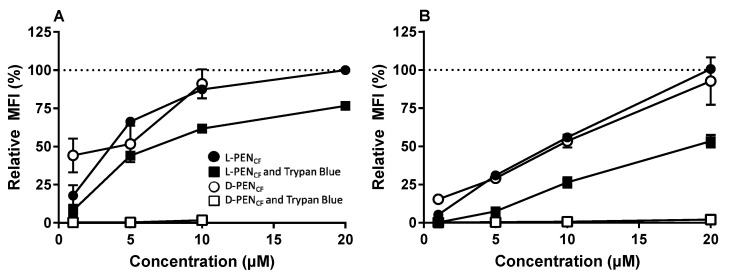
Relative uptake of l-PEN_CF_ and d-PEN_CF_ into HepG2 cells (**A**) or Caco-2 cells (**B**). 1–20 µM CPP in hHBSS (pH 7.4) was incubated with cells. l-PEN_CF_ (black circle), l-PEN_CF_ followed by 0.1% (*v*/*v*) Trypan Blue (black square), d-PEN_CF_ (white circle), and d-PEN_CF_ followed by 0.1% (*v*/*v*) Trypan Blue (white square). Data are relative to those of cells incubated with 20 µM l-PEN_CF_ (indicated by dotted line). Mean ± SEM, N = 2–3, *n* = 3.

**Figure 4 pharmaceutics-15-01672-f004:**
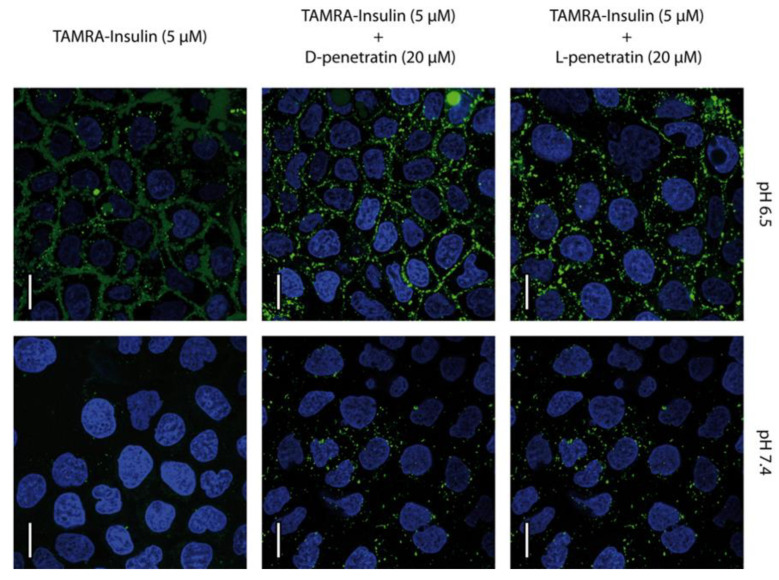
PEN-mediated delivery of TAMRA-insulin into Caco-2 cells at pH 6.5 and 7.4. Cells were incubated with TAMRA-insulin (5 µM, green) and l-PEN or d-PEN (20 µM) for 1 h, followed by staining of nuclei with Hoechst 33,342 (blue), while no membrane-compromised cells were found upon including staining with DRAQ7 as no magenta DRAQ7 fluorescence was detected. The images are representative of the same passage of cells, and the experiment was repeated in two different passages of cells. Scale bar 20 µm.

**Figure 5 pharmaceutics-15-01672-f005:**
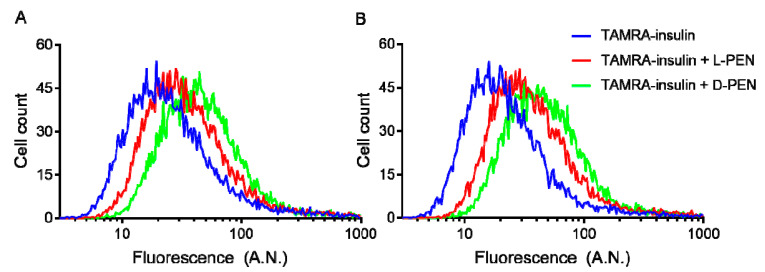
Uptake of TAMRA-insulin in Caco-2 cells displayed as histograms at pH 6.5 (**A**) and 7.4 (**B**). Cells were treated with test solution (5 µM TAMRA-insulin only (blue) as well as with 20 µM l-PEN (red) or d-PEN (green)) for 1 h, followed by trypsinization and analysis by flow cytometry. Mean fluorescence in arbitrary units (A.N.) from two independent experiments.

**Figure 6 pharmaceutics-15-01672-f006:**
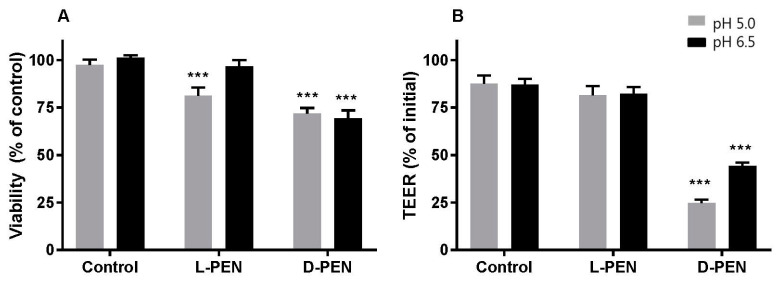
Viability (**A**) as indicated by metabolic activity using the MTS/PMS assay, and integrity (**B**) measured by TEER of Caco-2 epithelium upon exposure to l-PEN or d-PEN at 40 µM at pH 5 (grey) and pH 6.5 (black). Data were pooled from permeation experiments performed separately for each cargo. Metabolic activity was calculated relative to the activity of buffer-exposed control cell monolayers at pH 6.5. Asterisks indicate significant differences (*** *p* < 0.001). Mean ± SEM, N = 2–3, *n* = 12.

**Figure 7 pharmaceutics-15-01672-f007:**
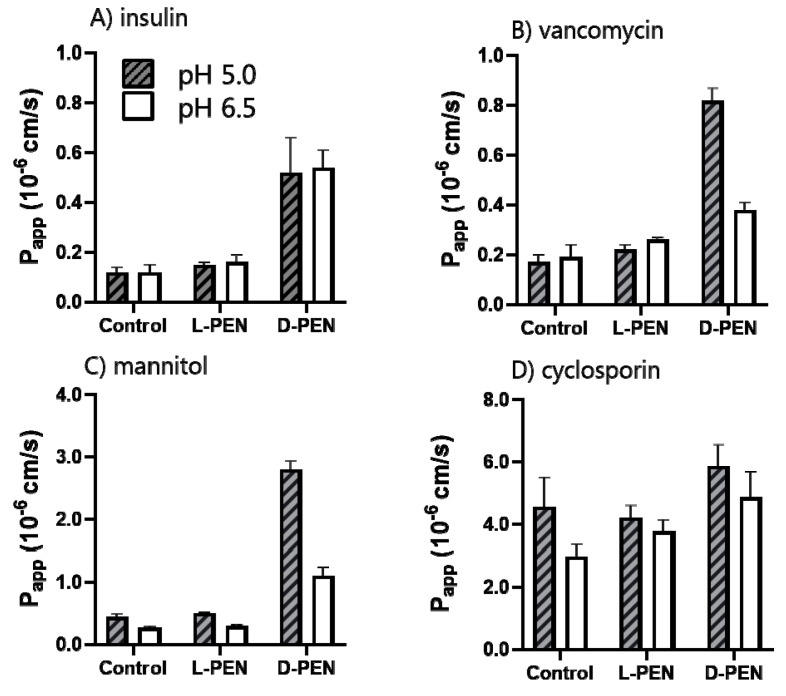
Apparent permeability coefficients (P_app_) for different cargos across Caco-2 monolayers in the presence of 40 µM l-PEN or d-PEN at pH 5.0 (grey, dash) and 6.5 (white). TAMRA-insulin (**A**), vancomycin (**B**), mannitol, and (**C**) cyclosporin (**D**). Mean ± SEM, N = 2–3, *n* = 3.

**Table 1 pharmaceutics-15-01672-t001:** Test compounds were included for investigating the delivery capacities of PEN enantiomers.

Compound	Mass (Da)	Log P	Note
Insulin	5808	−13.1	Hydrophilic two-chain peptide
Mannitol	182	−3.7	Sugar-derived polyol (paracellular marker)
Vancomycin	1449	−4.4	Hydrophilic tricyclic glycopeptide
Cyclosporin	1202	3.6	Hydrophobic monocyclic peptide

Physicochemical properties (mass and predicted log P) were obtained from Drugbank.com and pubchem.ncbi.nlm.nih.gov (accessed on 3 January 2023).

**Table 2 pharmaceutics-15-01672-t002:** Uptake of TAMRA-insulin (5 µM) in Caco-2 cells incubated at pH 6.5 and 7.4 alone or in the presence of 20 µM l-PEN or d-PEN for 1 h, followed by trypsinization, and analysis by flow cytometry in the presence or absence of Trypan Blue.

	pH 6.5		pH 7.4	
	Relative Uptake (%)	MFI (A.N.)	Relative Uptake (%)	MFI (A.N.)
**TAMRA-Insulin**				
Without Trypan Blue	100.0 ± 1.6	48.0 ± 1.2	100.0 ± 2.2	41.2 ± 2.9
With Trypan Blue	103.4 ± 5.6	49.6 ± 1.9	100.8 ± 4.3	40.8 ± 2.6
**TAMRA-Insulin + l-PEN**				
Without Trypan Blue	126.5 ± 7.7	60.4 ± 2.6	152.6 ± 6.9	61.9 ± 1.5 ***
With Trypan Blue	124.7 ± 7.3	62.3 ± 5.1	151.6 ± 2.8	62.1 ± 4.5 ***
**TAMRA-Insulin + d-PEN**				
Without Trypan Blue	148.2 ± 2.0	71.1 ± 1.4	181.1 ± 15.3	72.6 ± 1.6
With Trypan Blue	120.0 ± 7.1	59.1 ± 2.0 *	164.6 ± 10.5	65.9 ± 1.0

Mean fluorescence in arbitrary numbers (A.N.) from two independent experiments. Asterisks indicate significant differences (* *p* < 0.05; *** *p* < 0.001). Mean ± SD, N = 2, *n* = 3.

## Data Availability

Data are available upon reasonable request.

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
