# Peer review of "Stereoisomer-Dependent Membrane Association and Capacity for Insulin Delivery Facilitated by Penetratin"

_pharmaceutics, 2023, doi:10.3390/pharmaceutics15061672_

Round 1

Reviewer 1 Report

The article by Birch et al. reports on the cellular uptake and drug delivery ability of penetratin steroisomers. The manuscript is very well written and reader-friendly. The results are relevant in the field and I am happy to recommend the publication of this work in Pharmaceutics.

I would suggest the authors to discuss the protease stability of penetratin steroisomes. This is introduced in the abstract and mentioned in the article, but it would be good to better highlight this crucial feature citing relevant papers in which the proteolytic stability of peptides composed of d-aminoacids is discussed.

Author Response

The authors appreciate the suggestion to include further information on the protease stability of the stereoisomers and thereby extend the discussion towards an in vivo evaluation of the stereoisomers. To accommodate this, we have added to the discussion with further reference to the already included reference 33 (Nielsen et al., J Control Release 189 (2014) 19–24) and reference 29 (Birch et al., Biochem J (2018) 475 (10): 1773–1788). Further, the reference Kremsmayr et al., J. Med. Chem. 2022, 65, 6191−6206 is included in the discussion of the revised version of the manuscript as reference.

Reviewer 2 Report

In this paper Birch et al. describe the comparison of L- and D-penetratin in the delivery of inzulin in cellular models and also other test compounds in transepithelial delivery. As the mechanism of D-penetratin is not yet well described, and differs from cell to cell and cargo to cargo, this work provides a valuable contribution to this field. The manuscript is well written, the experiments are adequately described.

I recommend publication in Pharmaceutics.

Author Response

The authors appreciate the reviewers´ positive opinion about the work and manuscript.

Reviewer 3 Report

The authors aimed to compare membrane association, cellular uptake, and delivery capacity for all-L and all-D enantiomers of penetratin (PEN) by using different cell models and cargos. They claimed that the stereochemistry of PEN influenced the transepithelial delivery propensity, but not the cellular uptake. However, the evidence is too weak and the protocol was designed badly, which cannot support the conclusion. Therefore, I recommend to reject the manuscript to be published in this journal.

Major review:

1. Why the authors used the cancer cells, such as Caco-2 and HepG2 cells to evaluate the cellular uptake of insulin. I am really confused on its significance.

2. The capability of CPP enantiomers will show little effect on the oral delivery for the complex gastrointestinal environment. The potential for the oral delivery of macromolecular therapeutics was out of the question.

3. There is no any information to confirm the delivery by CPP.

4. The abstract was written badly.

Minor revisions:

1.    Figure 3, no legend was showed to each line. The same problems in Figure 5. Similarly, figure 6 and figure 7 should be also illustrated by legend.

2.    Figure 4, the fluorescence intensity even the DAPI labeled nucleic in the first image was inconsistent with that of the latter two images.

3.    The discussion and conclusion sections seem overquoted and the thesis is not clear enough.

Author Response

Comment 1:

In response to this comment, it is well known that the Caco-2 cell epithelial model constitute a frequently used (academia and industry) and thus valuable model for in vitro assessment of oral delivery of small molecule drugs as well as macromolecule drug delivery systems (e.g. references 3, 28, 40 in the manuscript, and Mehta et al., 2023).

Mehta et al., Drug Permeability - Best Practices for Biopharmaceutics Classification System (BCS)-Based Biowaivers: A Workshop Summary Report, J. Pharm. Sci. 2023, in press doi10.1016/j.xphs.2023.04.016

HepG2 cells are used in the study as model hepatocytes in line with other frequent reports (e.g. Roursgaard et al.,2016; Thomes et al., 2023). In this study, we use HepG2 only to investigate the uptake of the CPPs to demonstrate their interactions with different types of cell membranes, and thus not to evaluate the delivery of insulin. This is a misunderstanding by the reviewer. We find that the manuscript in its current form clearly validates our use of the HepG2 cells.

Roursgaard et al., In vitro toxicity of cationic micelles and liposomes in cultured human hepatocyte (HepG2) and lung epithelial (A549) cell lines, Toxicology In Vitro 36 (2016) 164-171. doi: 10.1016/j.tiv.2016.08.002.

Thomes et al., Ethanol Exposure to Ethanol-Oxidizing HEPG2 Cells Induces Intracellular Protein Aggregation, Cells 12 (2023) 1013. doi: 10.3390/cells12071013.

Comment 2:

References 30 and 33 in the manuscript report on the use of enantiomers as delivery enhancing agents, thus we respectfully disagree with the reviewer on this point.

Comment 3:

We are finding it very difficult to understand the point raised here. However, the experiments are based on assessing the delivery capacity of the peptides studied here versus insulin alone. Thus, our findings clearly demonstrate delivery is mediated to different extents by the studied CPPs.

Comment 4: 

The abstract was written according to the style required by the journal. We have looked at this again and find that it is a clear representation of the motivation for the work, the experiential aims, the findings, and the conclusions.

Minor comment 1:

We have accommodated the reviewers´ suggestion and included legends directly in the figure, and not only in the figure text.

Minor comment 2:

Yes, there were very clear differences in both the intensity and the cellular distribution of the peptides and this is a major finding for this experiment. We do note that the nucleus is more faint in the Insulin pH 6.5 control but the intensity still allows for us to perform analysis of the cellular location of the peptide, that were essentially interested in. Please note that all images in this figure were generated using the same microscopy and processing settings.

Minor comment 3:

We are not sure we understand this point raised by the reviewer but amendments have been made in the Discussion to address a point made by another referee.

Reviewer 4 Report

Accept in present form.

Author Response

(The authors gave the same response as above.)

Round 2

Reviewer 3 Report

The authors seemed to address all of the issues.